# Can Teledentistry Replace Conventional Clinical Follow-Up Care for Minor Dental Surgery? A Prospective Randomized Clinical Trial

**DOI:** 10.3390/ijerph19063444

**Published:** 2022-03-15

**Authors:** Diana Heimes, Philipp Luhrenberg, Nils Langguth, Sebahat Kaya, Christine Obst, Peer W. Kämmerer

**Affiliations:** 1Department of Oral- and Maxillofacial Surgery, University Medical Center Mainz, 55131 Mainz, Germany; philipp.luhrenberg@unimedizin-mainz.de (P.L.); nils.langguth@unimedizin-mainz.de (N.L.); sebahat.kaya@unimedizin-mainz.de (S.K.); peer.kaemmerer@unimedizin-mainz.de (P.W.K.); 2Private Practice, In den Weingärten 4, 67551 Worms, Germany; christine-obst@hotmail.com

**Keywords:** teledentistry, telemedicine, oral health, follow-up, oral surgery

## Abstract

(1) Background: Born out of necessity, the implementation of digital processes experienced significant increase during the COVID-19 pandemic. Here, telemedicine offered a bridge to care and now an opportunity to reinvent virtual and hybrid care models, with the goal of improved healthcare access, outcomes, and affordability. The aim of this monocentric prospective, randomized trial was to compare conventional to telephone follow-up after minor dentoalveolar surgery on the basis of special aftercare questionnaires. (2) Methods: Sixty patients who underwent dentoalveolar surgery under local anesthesia were randomly assigned to both groups. After an average of four days, either telephone follow-up (test) or conventional personal aftercare (control) was performed. Based on the questionnaire, the following subject areas were evaluated: symptoms, complications, satisfaction with practitioner, travel, and waiting time, as well as the preferred form of follow-up care. (3) Results: There was no statistically significant difference regarding frequency of symptoms or complication rate. Patients who were assigned to the test group showed a clear tendency to prefer telephone follow-up (83.3%) to conventional aftercare (16.7%, *p* = 0.047). (4) Conclusions: The data suggest high acceptance of telephone-only follow-up after dentoalveolar surgery. The implementation of telemedicine could be a time- and money-saving alternative for both patients and healthcare professionals and provide healthcare access regardless of time and space.

## 1. Introduction

Dentoalveolar procedures are commonly performed without complications; even after surgery, patients only seldom experience severe problems. From this perspective, and in light of the current pandemic situation, which emphasizes minimizing the number of contacts to protect individuals and vulnerable groups, the benefit of traditional clinical follow-up care may be questioned [1].

Telemedicine (TM) literally means “healing from a distance.” In the 1970s, this term was coined by Thomas Bird, but origins of TM can be traced back to the early 20th century. Since then, various definitions of TM have been implemented, highlighting a constantly evolving field that is becoming increasingly important due to technological advances. According to the World Health Organization (WHO), TM is defined as “The delivery of health care services, where distance is a critical factor, by all health care professionals using information and communication technologies for the exchange of valid information for diagnosis, treatment and prevention of disease and injuries, research and evaluation, and for the continuing education of health care providers, all in the interests of advancing the health of individuals and their communities” [2].

TM is meant to (1) provide clinical support, (2) overcome geographical barriers, (3) involve the use of various types of information and communication technologies, and aims for (4) improving health outcomes [3].

Postoperative follow-up via telephone consultation has been performed in various disciplines since the 1990s [4]. Thereby, the need for clinical consultation can be recognized in time, and intervention can be made accordingly [5,6,7,8].

A key condition for the development of TM in Germany is the “Act for Secure Digital Communication and Applications in Healthcare” (E-Health Act), which entered into force in 2016 [9,10]. However, the Medical Association’s professional code of conduct for physicians practicing in Germany prohibited exclusive telecare [11]. The amendment of this section in 2018 allows the support of face-to-face treatment through the use of communication technologies [12].

The current pandemic situation in particular is making alternatives to traditional clinical follow-up care increasingly important. Nevertheless, not only infectious diseases speak for an expansion of telemedical care [13]. Alongside reducing travel and waiting times for patients—in rural areas in particular, it is often difficult to provide health care due to long travel distances [14,15,16]—the additional use of technical aids such as telephone and internet to provide follow-up care could significantly reduce the workload for healthcare professionals [14,15].

According to Gray et al., postoperative clinical follow-up is expected by patients even after routine surgery [16], but demands for efficient, accessible, resource-saving, and, at the same time, patient-oriented and safe service delivery necessitate changes and the establishment of alternative concepts [17]. Molfenter et al. showed that the implementation of TM made patients feel more integrated in their treatment and that they had the feeling of being better monitored by their physician [18]. It is also expected that diagnoses will be secured more quickly or that the number of misdiagnoses will be reduced due to easier access to medical expertise and specialized services, regardless of time and space [19,20]. On the other hand, the inability to perform a detailed physical examination of the patient, the lack of nonverbal information, and the risk of a deterioration of the doctor–patient relationship are disadvantages of a TM-only follow-up [21,22,23,24,25,26,27].

The lack of data regarding clinical and financial benefits [20] and patient satisfaction [21] is problematic. Available data from randomized trials of telehealth applications vary, and further research and evaluation is needed before recommendations are made to invest in and increase the use of technologies that have not been sufficiently tested [22]. Legal and ethical aspects such as the use of confidential patient data, liability issues, and responsibilities also remain to be clarified [19,24,25,26,27]. Further comparative studies are required, particularly in oral and maxillofacial surgery and dentistry [14]. Therefore, this study aims to investigate patient-related outcomes after dentoalveolar surgery under traditional clinical follow-up compared to those under telemedical care.

## 2. Materials and Methods

### 2.1. Patients

Sixty-eight patients treated at the Department of Oral and Maxillofacial Surgery at the University Medical Center Mainz over an eight-month period from May to December 2020 were included within this study after approval of the local ethics committee (No. 2019-14472-NIS). Patients were informed about the study during consultation and planning of the dentoalveolar treatment. If interested, a detailed explanation was given; information about the study was handed out in written form and informed consent was obtained. All patients were equally informed about the postoperative behavior such as physical rest and soft diet.

### 2.2. Inclusion and Exclusion Criteria

Patients who underwent dentoalveolar surgery (tooth extraction or osteotomy) under local anesthesia at the Department of Oral and Maxillofacial Surgery at the University Medical Center Mainz were included in the study. The detailed inclusion and exclusion criteria are listed in Table 1.

### 2.3. Study Design

The study was designed as a monocentric prospective, randomized trial. All procedures were performed according to the hygienic and surgical standards by experienced maxillofacial surgeons and dentists of the Department of Oral and Maxillofacial Surgery at the University Medical Center Mainz. After the procedures, the extractions sockets/osteotomy wounds were debrided thoroughly and sometimes sutured with resorbable suturing material. Patients were instructed to bite on cotton swabs compressing the extraction alveolus to prevent them from bleeding. They received general instructions on how to act after surgery (removal of the swabs after thirty minutes, cooling from extraoral with ice packs for two days, taking pain killers if required, eating liquid food for a few days, no smoking).

Once the surgical procedure was successfully performed, the patients were assigned to the experimental group (follow-up by telephone) or control group (follow-up in person) according to a randomization list generated using Microsoft Excel^®^ software (Figure 1). After an average of four days, a telephone call or personal appointment was scheduled.

### 2.4. Questionnaires

A case report form (CRF) with standardized questions was developed for postoperative TM follow-up and for personal follow-up. The CRF included choice questions and open questions (Table 2) aiming to ensure freedom from symptoms, identify possible postoperative complications, and evaluate the practitioner and the form of follow-up. The questions were developed based on an expert consensus of experienced oral and maxillofacial surgeons and are based on current guidelines. In case of personal follow-up, travel distance and time, as well as waiting time, were also recorded.

The physician then assessed the need for a personal follow-up care and noted the time needed for the appointment (minutes). Comments were noted in a special field.

### 2.5. Statistics

Statistics were performed using IBM SPSS Statistics for Macintosh, version 27 (Armonk, NY, USA, IBM Corp). To analyze the differences between the measured values, normality (Kolmogorov–Smirnov) and homogeneity of variance tests (Levene statistic) were performed at first in order to check the conditions for the subsequent analysis. The *p*-values were obtained with an independent samples t-test. In case of not normally distributed values, a Mann–Whitney test was used instead. A *p*-value < 0.05 was termed significant.

## 3. Results

### 3.1. Study Population

The mean age of the patients included within this study was 51.6 ± 18.6 years. Overall, 45.6% of subjects were male and 54.4% were female. Study participants in the experimental group had a median age of 54 years, while participants in the control group had a median age of 49 years (*p* = 0.208). Female patients were more frequent within the experimental group (60.6%); male patients made up 51.4% of the control group.

The average distance between the residence of the patients and the clinic was 21.7 km without differences between groups (*p* = 0.598). In the experimental group, a minimum travel distance of 1.2 km and a maximum travel distance of 122.0 km were measured for which travel times between four minutes and 1.45 h were calculated. The travel distance of patients within the control group was 2–119 km. This resulted in travel times of six minutes to 1.4 h (*p* = 0.864) (Table 3).

With 60.3% of the surgeries performed, simple tooth extractions represented the largest proportion of procedures included within this study (experimental group: 63.6%, control group: 57.1%, *p* = 0.587). Wisdom tooth extractions were the second most common procedure (25.0% in total; experimental group: 27.3%, control group: 22.9%, *p* = 0.677). Osteotomies of other teeth were performed in 14.7% of cases in total (experimental group: 9.1%, control group: 14.7%, *p* = 0.208). A total of 129 teeth were treated; 83 teeth were extracted, 22 teeth were osteotomized and 24 wisdom teeth were removed. Between one and 13 teeth were removed per patient (M = 2 ± 0.2). No statistically significant differences were seen between the groups (*p* = 0.910). Within the experimental group, 66.7% of the cases, and in 74.3% of the cases in the control group, the surgical wound was closed by resorbable sutures (*p* = 0.494) (Appendix A).

### 3.2. Freedom of Symptoms

A total of 3.3% of the study participants reported feeling sick on the day of follow-up (experimental group: 6.7%, control group: 0%, *p* = 0.154). A total of 16.7% of participants took pain medication in the form of non-steroidal anti-inflammatory drugs (16.7% each, *p* = 1.00). Ten percent of the patients reported of limitations in performing daily activities such as lifting heavy objects (10% each, *p* = 1.00), and in 8.3% of the cases they expressed concerns (experimental group: 13.3%, control group: 3.3%, *p* = 0.165). A proportion of 43.3% reported continuing to eat liquid–soft food (experimental group: 36.7%, control group: 50%, *p* = 0.301) (Figure 2, Appendix A).

### 3.3. Exclusion of Complications

In 1.7% of cases, patients reported a taste of blood (experimental group: 0%, control group: 3.3%, *p* = 0.317). A proportion of 5% experienced swelling of the surgical area (experimental group: 3.3%, control group: 6.7%, *p* = 0.557). Fever, chills with sweat, and dysphagia or difficulty breathing were not reported (*p* = 1.00) (Figure 3, Appendix A).

### 3.4. Evaluation of the Practitioner

The whole patient collective was satisfied with their practitioner (*p* = 1.00). In 25% of the cases, there were questions concerning the discharge interview and postoperative follow-up, which were subsequently answered (experimental group: 33.3%, control group: 16.7%, *p* = 0.139). The questions mainly concerned the reabsorption time of the suture material and the further course of therapy (Figure 4, Appendix A).

### 3.5. Evaluation of the Form of Follow-Up Care

Within the experimental group 16.7% of the participants and 40% of the control group reported the wish for a personal follow-up appointment (*p* = *0*.047).

On average, patients waited 12.8 min for their appointment; maximum waiting time was 45 min. This time was rated as a long waiting time by 20% of the study participants. Within the experimental group, average phone calls lasted 3.9 min (±2.1), and participants within the control group had a follow-up check-up of a mean of 4.7 min (±3.9) (*p* = 0.178) (Figure 5, Appendix A).

### 3.6. Quality of the Form of Follow-Up Care

Within the experimental group in two cases (6.7%), personal follow-up care was considered based on the telephone conversation. Patients reported feeling sick, taking pain medication, being limited in performing activities of daily living, and having concerns. During the further clinical examination, wound-healing disorders with persistent pain were diagnosed. Within the control group, follow-up care was considered necessary in three cases (10%). Patients reported continuing to take pain medication; in two cases, performing activities of daily live was limited, and in one case each, concerns were expressed or a taste of blood in the mouth was reported. During the clinical examination, two wound-healing disorders and one surgical site infection were noted. Statistical analysis showed no significant difference between the two groups (*p* = 0.643) (Appendix A).

## 4. Discussion

As early as 1998, Wootton et al. suggested that utilizing TM will lead to a restructuring of health care delivery, particularly in outpatient care, medical education, and management meetings [28]. The ongoing COVID-19 pandemic provided an advance in global digitization and requires a tremendous effort from healthcare systems and rapid adaptation [29]. Digital technologies such as TM are essential to the availability of healthcare during the pandemic [30] and are able to reduce the risk of infection with SARS-CoV-2 for both patients and healthcare workers [31]. With the increasing digitization not only in medicine, but in all areas of life, it is quite conceivable that in the future, a broader acceptance will be present in the general population. The development of new techniques in medicine offers numerous advantages over conventional concepts. In view of an aging population, the number of physician consultations will continue to increase. Given this development, and of course in the face of potential further pandemic events, the development of new, time- and cost-saving alternatives to face-to-face perioperative management of the patient is of paramount importance. Here, TM can address the healthcare disparity that exists between patient needs and healthcare availability.

The COVID-19 pandemic forced the medical system to find ways to safely access and deliver healthcare. In April 2020, overall TM utilization was measured to be 78 times higher than before the pandemic. This change was enabled by (1) an increased willingness of both healthcare professionals and patients to use TM, and (2) regulatory changes. During the COVID-19 pandemic, TM offered a bridge to deliver medical care over distance, and now offers a chance to improved healthcare access, outcomes, and affordability [32].

This is particularly advantageous for patient presentations that involve time-consuming transportation due to infections with multiresistant bacteria, being bedridden, or need for nursing care [33], as well as for chronically ill patients, whose quality of life can be improved due to saved travel and waiting times [34,35,36]. In general, a faster confirmation of diagnoses or a lower number of misdiagnoses due to easier access to medical expertise and specialized services, regardless of time and space, is highlighted in the literature as another factor supporting expansion of TM care [19,20], whereas, in particular, the lack of technical competence of participants—especially older patients—could further complicate the implementation of TM services [37].

Studies concerning postoperative TM follow-up in dentistry have been conducted in the UK, Italy, and the USA. The studies described a high overall acceptance of TM follow-up care with a high level of patient satisfaction. In none of the studies was the rate of postoperative complications higher in the experimental group than in the group with conventional face-to-face follow-up. In consideration of the numerous advantages and analogue to the findings of the current study, a large proportion of patients subsequently preferred TM to face-to-face follow-up [1,26,36].

The optimal time for a telephone control appointment remains unclear [22]. Pain has been reported in the literature as the most common symptom [38] and key factor [39] in postoperative care after dental procedures. If the intensity does not decrease within the first few days, it may indicate a complication present [40,41]. Experience has shown that most of the pain subsides after the third postoperative day. Swelling occurs between 12 and 24 h and reaches its maximum after 48–72 h [42,43,44]. Thus, the postoperative follow-up was performed on average on the fourth postoperative day within this study.

Five of the 60 cases analyzed (8.3%) showed postoperative complications at the time of follow-up. This percentage is below the data in comparable studies. Susarla et al. diagnosed postoperative complications in 19.5% of cases [1], and in the randomized clinical trial conducted by Pippi et al., complication rates of up to 30.7% were registered [38]. Here, it must be taken into account that dentoalveolar procedures are performed in a non-sterile surgical field. The low complication rate of the present study might be due to the inclusion of patients with a low risk of complications. On the basis of the study performed, TM appears to be a useful, efficient, and positively evaluated technical advancement of the medical sector, although some limitations should be considered. If there is an increased risk of postoperative complications, telemedical follow-up alone should be questioned on the basis of current data. Risk groups include patients with acute and/or chronic infections in the surgical area [45,46,47], higher age [40,47,48,49], incompliance [47], impacted teeth [45,46,50,51,52], poor oral hygiene [45,53], systemic diseases affecting wound healing (chronic renal or liver diseases, diabetes mellitus, immunosuppression, malnutrition) [47,54], prolonged operation time [43,48,49,51], history of radiation, chemotherapy or antiresorptive medication [47,55], bleeding disorders (e.g., hemophilia, thrombocytopenia), antiplatelet therapy or anticoagulation [55], and habits such as smoking and drinking [47,56].

A proportion of 71.7% of the total study population preferred TM follow-up to a face-to-face appointment. The difference in values between the experimental and control groups is of significant importance. While 83.3% of the patients in the experimental group preferred the TM follow-up, only 60% of the patients in the control group expressed this preference. This statistically significant difference suggests that satisfaction with telephone follow-up is increased by experience with this form of aftercare.

The present results are comparable to those of Pippi et al., Ainsworth et al., and Susarla et al., who reported patient satisfaction of 73% [24], 86% [38], and 90% [1].

A small difference regarding the length of postoperative follow-up between the experimental and control group were noticed. If travel and waiting time of the conventional follow-up is added, the time spent for aftercare differs by an average of 1.09 h. Clinical processes for preparation and post-processing of the treatment room for personal aftercare were not calculated in the study but should be taken into account when evaluating the time efficiency of telemedical follow-up.

With an average distance of 21.7 km from the patient’s place of residence to the clinic, travel costs of EUR 10.84 were calculated (fuel costs and running costs). With an average waiting time of 12.8 min and treatment duration of 4.7 min, parking costs of EUR 0.80 resulted. The maximum distance of 122 km results in travel costs of EUR 60.94. Parking costs can be as high as EUR 2.80 for the longest registered waiting time and treatment duration. Travel by public transport was not evaluated within this study.

Thus, anticipated benefits of TM applications include time-saving, both patient- and clinic-related cost-savings, improved accessibility of medical care for patients independent of their location, more patient comfort or quality of life—especially for patients who need regular check-ups—and the reduction of unnecessary face-to-face appointments [8,16,17,22,27,57,58,59,60,61,62,63,64]. On the other side, nonverbal communication is difficult in TM care which in turn could deteriorate the doctor–patient relationship and complicate the diagnosis of disease patterns that are not reported explicitly. Furthermore, the lack of physical examination could lead to the non-detection of pathologies. The need of self-assessment could further overstrain the patients’ abilities.

Limitations of this study are the rather small patient collective and, on the other hand, the use of only one method of information and communication technology (telephone). It should be noted that a substantial proportion of 83.6% of the 710 patients in total could not be included in the study due to the inclusion and exclusion criteria. This is because tooth extractions are mostly performed in older patients, who are more likely to have comorbidities [20]. As a referral center for general practitioners and dentists, a university hospital more frequently treats patients who have a particularly high-risk profile for intraoperative and postoperative complications due to various pre-existing conditions.

In addition to the telephone follow-up examined in this study, other formats would also be conceivable. These include regular check-ups of chronically ill patients and perioperative management including patient education and follow-up care, as well as the appropriate triage of patients. All in all, the expansion of TM not only allows faster distance-independent access to medical services, but also makes it possible to strengthen the patients’ self-responsibility for their disease by improving their understanding of it. This can be a major step toward strengthening preventive, over therapeutic, measures [65].

An expansion of the follow-up via video communication would be possible and useful, as it would allow objective assessment of the patient and wound situation in addition to subjective patient reports.

Further research is needed, especially in sectors with a high number of outpatient surgeries such as maxillofacial and dental medicine. To date, the number of studies focusing on the effectiveness of TM follow-up examinations is very limited in dentistry. In terms of improved healthcare access, better outcomes, and affordability, a wide range of information and communication technologies have to be investigated. In addition to the method of contacting the patient, it is still necessary to identify certain risk groups for whom TM follow-up alone cannot be recommended. Likewise, the possible areas of use for digital applications should be expanded and analyzed in studies to ensure the broadest possible use to improve patient care.

## 5. Conclusions

The data collected in the present study suggest a high acceptance and safety of telephone-only follow-up after dentoalveolar surgery in the studied population. The implementation of telemedical examination in everyday clinical practice could be a time- and money-saving alternative for both patients and healthcare professionals and provide healthcare access regardless of time and space. Further research is needed in the use of a wider range of information and communication technologies as well as different areas of patient care.

## Figures and Tables

**Figure 1 ijerph-19-03444-f001:**
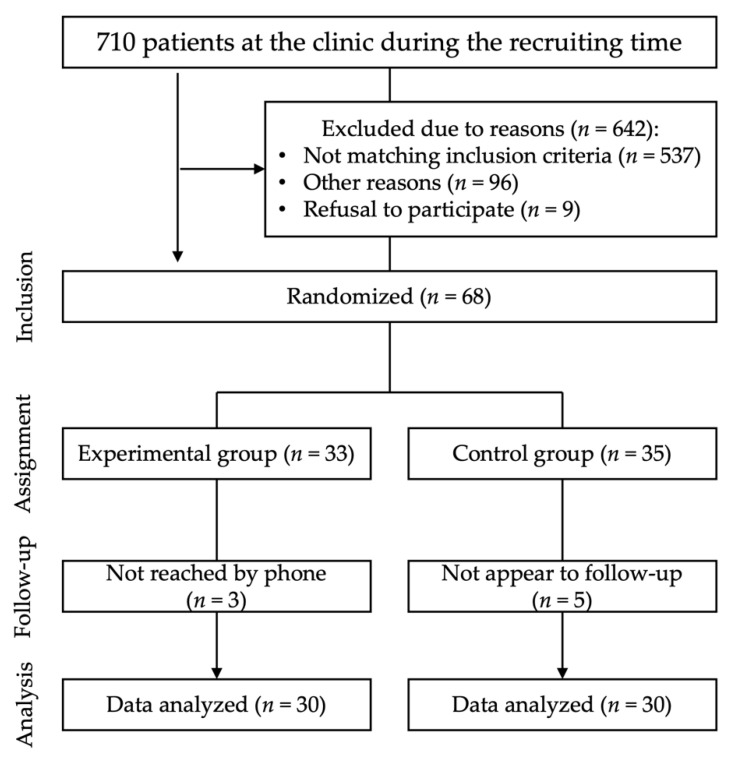
Assignment to either telephone or clinical follow-up group. Patients excluded due to reasons either did not match the inclusion criteria (no minor surgery as defined by tooth extraction or osteotomy), were excluded because of refusal to participate, or due to other reasons such as matching the exclusion criteria or complications during surgery such as severe bleeding or swelling.

**Figure 2 ijerph-19-03444-f002:**
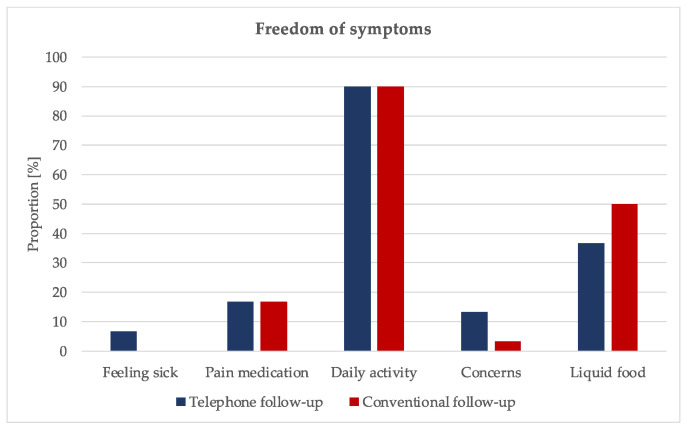
Bar charts illustrating freedom of symptoms classification between the two groups.

**Figure 3 ijerph-19-03444-f003:**
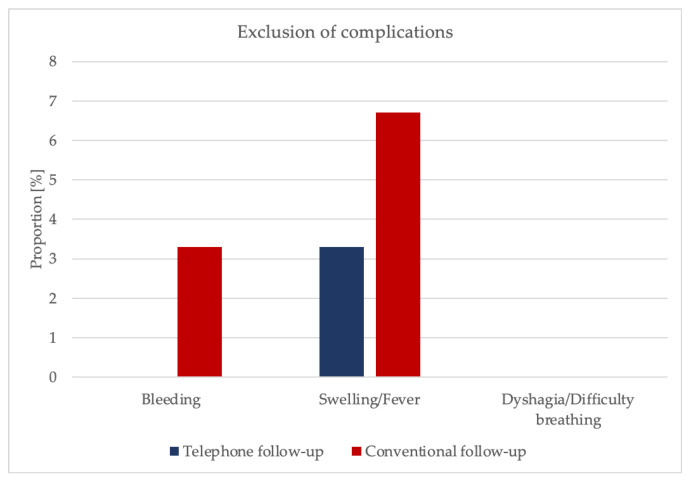
Bar charts illustrating exclusion of complications.

**Figure 4 ijerph-19-03444-f004:**
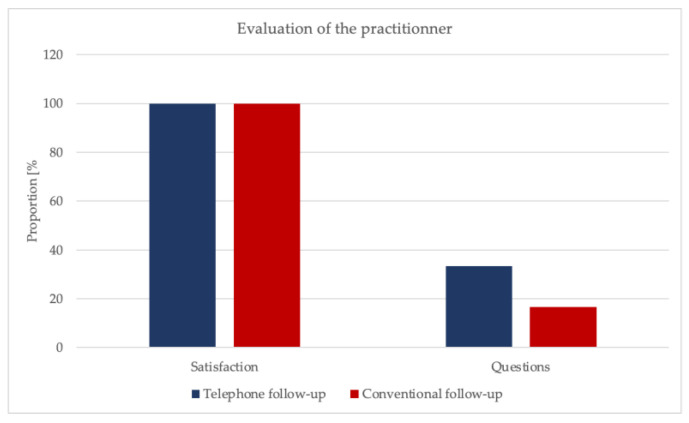
Evaluation of the practitioner.

**Figure 5 ijerph-19-03444-f005:**
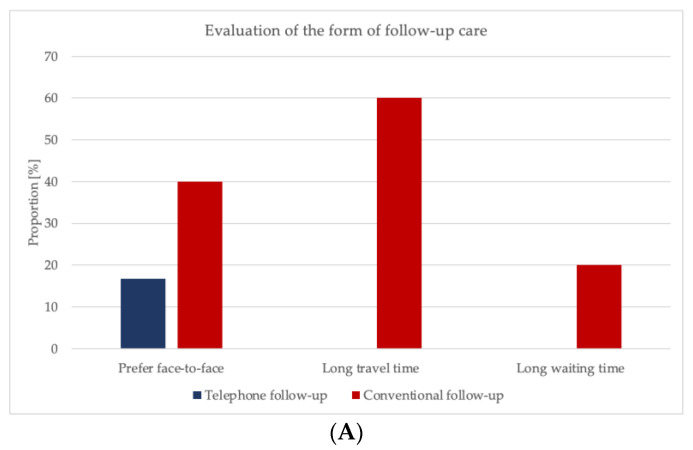
(**A**) Evaluation of the form of follow-up care. (**B**) Duration of the appointment.

**Table 1 ijerph-19-03444-t001:** Inclusion and exclusion criteria for study participation.

Inclusion Criteria	Exclusion Criteria
Consent to study participation Tooth extractions and osteotomies under local anesthesia	Refusal to participate in the study Surgical procedure that requires flaps for closure (e.g., oro-antral perforation) Risk of post-operative bleeding (taking anticoagulants or antiplatelet agents) Risk for osteonecrosis of the jaw (history of radiation, chemotherapy or antiresorptive drugs) Immunosuppression Planned implantology treatment Age < 18 years Missing telephone line Language barrier

**Table 2 ijerph-19-03444-t002:** Questionnaire.

Class	Questions
Freedom of symptoms	Do you feel bad after surgery? Are you currently taking pain medication? Can you perform normal daily activities? Do you have any concerns? Do you continue to eat a liquid-soft diet?
Exclusion of complications	Do you have a taste of blood in your mouth or is there active bleeding? Do you suffer from swelling, fever, or chills with sweat? Is there any difficulty swallowing or breathing?
Practitioner	Are you satisfied with your surgeon? Are there any questions concerning the discharge consultation or the postoperative check-up?
Follow-up care	Would you either prefer telephone follow-up or personal follow-up
Travel and waiting time	Did you have a long travel time? Did you have a long waiting time?

**Table 3 ijerph-19-03444-t003:** Descriptive data of the study population.

	Telephone Follow-Up		Clinical Follow-Up		Total		
	Mean	SD	Mean	SD	Mean	SD	*p*
Age (y)	53.9	20.8	49.4	16.2	51.6	18.6	0.208
Travel distance (km)	23.1	25.2	20.3	22.4	21.7	23.7	0.598
Travel time (min)	26.4	17.5	25.1	16.0	25.7	16.6	0.864

## Data Availability

Data are available on request of the corresponding author.

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
