# Peer review of "Can Teledentistry Replace Conventional Clinical Follow-Up Care for Minor Dental Surgery? A Prospective Randomized Clinical Trial"

_ijerph, 2022, doi:10.3390/ijerph19063444_

Round 1

Reviewer 1 Report

I have reviewed the manuscript entitled "Can teledentistry replace conventional clinical follow-up care for minor dental surgery? A prospective randomized clinical trial".

The authors should answer the following questions:

- Were there no age limitations of the participants, and don't the authors think it would have been positive to delimit an age range in order to disregard very elderly patients?

- Figure 1 describes that 96 patients were eliminated from the study for "other reasons", what were these reasons?

- What was considered for minor dental surgery?

- Only extractions and osteotomies were analysed, and do the authors believe that the postoperative problems of both procedures are similar?

- Why were two study groups of 34 patients each not designed?

- Was the option of using a postoperative questionnaire with questions validated in previous studies evaluated?

- The conclusions are not correct. The authors should indicate in the conclusions the most salient points of their study, in summary form, and not include bibliographical references. The authors have described in the conclusions part of the discussion of their work.

- The bibliographical references are not described according to the journal's regulations.

Author Response

This is the revision of the manuscript entitled “Can teledentistry replace conventional clinical follow-up care for minor dental surgery? A prospective randomized clinical trial”.

We would like to thank the reviewer for the constructive comments and helping us to improve our work.

Comment 1: Were there no age limitations of the participants, and don't the authors think it would have been positive to delimit an age range in order to disregard very elderly patients?

Answer: Since a high age per se is not to be considered as a risk factor for surgery, but rather the general or medical condition of the patient is decisive, we decided against an upper age limit in this study. Due to the exclusion criteria listed, multimorbid or severely ill patients were excluded from participation in the study in general. Since this study addresses the interesting question of whether telephone follow-up is also sufficient, and we wanted to clarify this question for a population as broad as possible, we also decided against an upper aging limit. We hope that this explanation is sufficient for the reviewer. Otherwise, we would also like to address this in the section "study limitations".

Comment 2: Figure 1 describes that 96 patients were eliminated from the study for "other reasons", what were these reasons?

Answer: “Once the surgical procedure was successfully performed, the patients were assigned to the experimental group (follow-up by telephone) or control group (follow-up in person) according to a randomization list generated using Microsoft Excel® software (Figure 1). After an average of four days, a telephone call or personal appointment was scheduled.”

Other reasons were defined as follows: matching the exclusion criteria other than refusal to participate: the refusal to participate was particularly emphasized as a non-patient-related or -disease-related factor. Furthermore, change in surgical therapy, for example, expansion due to pro-antral perforation with need for flap formation or intraoperative complications, such as severe bleeding or swelling, were also among the other reasons mentioned that led to exclusion before randomization. Since randomization was performed after surgery, as mentioned above, to avoid unnecessary inclusion and exclusion for the above reasons, these were grouped under "other reasons".

This topic is now addressed in the caption of figure 1 as can be seen in lines 118–121.

Comment 3: What was considered for minor dental surgery?

Answer: Minor dental surgery was considered to be tooth extractions and osteotomies that do not require a flap for closure as defined in the inclusion and exclusion criteria.

Minor surgery was again explained in the caption of figure 1 as can be seen in lines 118–121.

Comment 4: Only extractions and osteotomies were analyzed, and do the authors believe that the postoperative problems of both procedures are similar?

Answer: We consider both procedures to be minor dental suergy for an oral/maxillofacial surgeon. In our opinion and experience, the two procedures can have very similar complications and side effects, although osteotomy must of course be considered the more invasive procedure. In order to distinguish clearly more invasive oral surgical procedures with an increased risk of complications, procedures requiring a flap (e.g. in the case of an oral-antral perforation or osteotomy of fully impacted wisdom teeth) were excluded.

Comment 5: Why were two study groups of 34 patients each not designed?

Answer: During the recruiting time, an equal distribution of the patients to the groups in question was planned. This would also have been carried out if numerous patients had not been excluded from participation by not fulfilling the inclusion criteria and if the recruitment period approved by the ethics committee had not ended. Since we had reached the number of statistically calculated necessary cases, there was no justification for extending the study, so we were satisfied with this minimal difference under the condition of a very similar number of cases.

Comment 6: Was the option of using a postoperative questionnaire with questions validated in previous studies evaluated?

Answer: The questions were developed based on an expert consensus of experienced oral and maxillofacial surgeons and are based on current guidelines such as the guideline on odontogenic infections of the German Society for Oral and Maxillofacial Surgery (DGMKG) and the German Society for Dental, Oral and Maxillofacial Medicine (DGZMK). This information was added in lines 126–128.

Comment 7: The conclusions are not correct. The authors should indicate in the conclusions the most salient points of their study, in summary form, and not include bibliographical references. The authors have described in the conclusions part of the discussion of their work.

Answer: This is correct. We moved the paragraph to lines 216–230 and 315–321 and added a new “Conclusion” as can be seen in lines 335–341.

Comment 8: The bibliographical references are not described according to the journal's regulations.

Answer: We are very sorry for this mistake and corrected it.

Reviewer 2 Report

Very interest manuscript concerning the pandemic era. Please correct the typo error in line 18

Author Response

This is the revision of the manuscript entitled “Can teledentistry replace conventional clinical follow-up care for minor dental surgery? A prospective randomized clinical trial”.

We are very thankful for the constructive comment and the help to improve our work.

Comment 1: Very interest manuscript concerning the pandemic era. Please correct the typo error in line 18.

Answer: We are very thankful for your comment and corrected the typo.

Reviewer 3 Report

The manuscript focuses on a very interesting and up-to-date. It is suitable for publications after the following concerns:

Page 1, line 18. Please revise “wo” (see: “Sixty Patients wo underwent dentoalveolar surgery”).

Page 4, Figure 1. Please explain the assignment to experimental (38) and control (35): why they were not allocated in equal numbers?

Page 4, line 120.  “A case report form (CRF) with standardized questions….” How was the questionnaire validated? Is the questionnaire taken from previous research? Please explain.

Page 5:

  • line 139. Please add standard deviation after the mean age.
  • “Overall, 45.6% of subjects were male 141 and 54.4% were female” – please move this sentence after the first one (“The mean age of the patients included within this study was 52 years”).

Page 8, lines 227-228. Please rephrase: “A few studies concerning postoperative TM follow-up in dentistry have been conducted in the UK, Italy and the USA”. It is not clear, in others countries have been conducted more studies?

Page 10, lines 308-330:

  • no bibliographic sources are added to the conclusions. Please redesign the conclusion part.
  • “As early as 1998, Wootton et al. suggested that utilizing TM will lead to a restructuring of health care delivery, particularly in outpatient care, medical education, and man agement meetings [16]”. It is not a conclusion of the present study, please remove or add to the discussion part.
  • “The ongoing COVID-19 pandemic provided an advance in global digitization and requires a tremendous effort by healthcare systems and rapid adaptation [63]”. The same mentioned suggestion as above.
  • “Digital technology such as TM are essential to the availability of healthcare during 313 the pandemic [64] and was able to reduce the risk of infection with SARS-CoV-2 for both 314 patients and healthcare workers [65]”. The same mentioned as above.
  • “With the increasing digitization not only in medicine, but in all areas of life, it is quite conceivable that in the future, a broader acceptance will be present in the general population. The development of new techniques in medicine offers numerous advantages over conventional concepts. In view of an aging population, the number of physician consultations will continue to increase”. Please mention the conclusions from your study, so rephrase the conclusion part.
  • Please remove reference numbers.

Best regards,

Author Response

This is the revision of the manuscript entitled “Can teledentistry replace conventional clinical follow-up care for minor dental surgery? A prospective randomized clinical trial”.

We would like to thank the reviewer for the constructive comments and helping us to improve our work.

Comment 1: Page 1, line 18. Please revise “wo” (see: “Sixty Patients wo underwent dentoalveolar surgery”).

Answer: Thank you. We corrected the typo.

Comment 2: Page 4, Figure 1. Please explain the assignment to experimental (38) and control (35): why they were not allocated in equal numbers?

Answer: During the recruiting time, an equal distribution of the patients to the groups in question was planned. This would also have been carried out if numerous patients had not been excluded from participation by not fulfilling the inclusion criteria and if the recruitment period approved by the ethics committee had not ended. Since we had reached the number of statistically calculated necessary cases, there was no justification for extending the study, so we were satisfied with this minimal difference under the condition of a very similar number of cases.

Comment 3: Page 4, line 120.  “A case report form (CRF) with standardized questions….” How was the questionnaire validated? Is the questionnaire taken from previous research? Please explain.

Answer: The questions were developed based on an expert consensus of experienced oral and maxillofacial surgeons and are based on current guidelines such as the guideline on odontogenic infections of the German Society for Oral and Maxillofacial Surgery (DGMKG) and the German Society for Dental, Oral and Maxillofacial Medicine (DGZMK). This information was added in lines 126–128.

Comment 4: Page 5, line 139. Please add standard deviation after the mean age.

Answer: We added a SD (line 143).

Comment 6: “Overall, 45.6% of subjects were male 141 and 54.4% were female” – please move this sentence after the first one (“The mean age of the patients included within this study was 52 years”).

Answer: We moved the sentence as suggested by the reviewer.

Comment 7: Page 8, lines 227-228. Please rephrase: “A few studies concerning postoperative TM follow-up in dentistry have been conducted in the UK, Italy and the USA”. It is not clear, in others countries have been conducted more studies?

Answer: The sentence has been rephrased as can be seen in lines 246 and 247.

Comment 8: Page 10, lines 308-330:

no bibliographic sources are added to the conclusions. Please redesign the conclusion part.

“As early as 1998, Wootton et al. suggested that utilizing TM will lead to a restructuring of health care delivery, particularly in outpatient care, medical education, and man agement meetings [16]”. It is not a conclusion of the present study, please remove or add to the discussion part.

“The ongoing COVID-19 pandemic provided an advance in global digitization and requires a tremendous effort by healthcare systems and rapid adaptation [63]”. The same mentioned suggestion as above.

“Digital technology such as TM are essential to the availability of healthcare during 313 the pandemic [64] and was able to reduce the risk of infection with SARS-CoV-2 for both 314 patients and healthcare workers [65]”. The same mentioned as above.

“With the increasing digitization not only in medicine, but in all areas of life, it is quite conceivable that in the future, a broader acceptance will be present in the general population. The development of new techniques in medicine offers numerous advantages over conventional concepts. In view of an aging population, the number of physician consultations will continue to increase”. Please mention the conclusions from your study, so rephrase the conclusion part.

Answer: Thank you for this helpful comment. We moved the paragraph to lines 216–230 and 315–321 and added a new “Conclusion” as can be seen in lines 335–341.

Round 2

Reviewer 1 Report

The authors have improved the manuscript. 

Reviewer 3 Report

Dear authors,

I agree with the publication of the manuscript in the present form, you adequately answered all the comments.

Best regards,